# Impact of 5' Adenosine Monophosphate, Potassium Chloride, and Glycine on the Physicochemical and Sensory Characteristics of Sodium-Reduced Chicken

**Jhunior Abrahan Marcía Fuentes** [1] , **Manuel de Jesús Álvarez Gil** [2], **Héctor Zumbado Fernández** [2],
**Ismael Montero-Fernández** [3] , **Daniel Martín-Vertedor** [4] , **Ajitesh Yadav** [5] **and Ricardo S. Aleman** [6,*]

1 Faculty of Technological Sciences, Universidad Nacional de Agricultura Road to Dulce Nombre de Culmi, Km 215, Barrio El Espino, Catacamas 16210, Honduras; jmarcia@unag.edu.hn
2 Facultad de Farmacia y Alimentos, Universidad de la Habana, Habana 10300, Cuba
3 Department of Chemical Engineering and Physical Chemistry, Area of Chemical Engineering, Faculty of Sciences, University of Extremadura, Avda. de Elvas, s/n, 06006 Badajoz, Spain
4 Technological Institute of Food and Agriculture CICYTEX-INTAEX, Junta of Extremadura, Avda. Adolfo Suárez, s/n, 06007 Badajoz, Spain
5 Department of Food Science and Human Nutrition, Iowa State University, Ames Iowa, IA 50011, USA
6 School of Nutrition and Food Sciences, Louisiana State University Agricultural Center, Baton Rouge, LA 70808, USA
* Correspondence: rsantosaleman@lsu.edu

**Abstract:** The demand for low-sodium products is growing worldwide and is compelled by the growing number of related illnesses. However, the quality of these products could be improved, likened to products produced with common salt (NaCL), because the replacement of sodium compromises the flavor of the product. Reducing sodium salts also poses an essential challenge for the meat industry, since sodium chloride (NaCl) fulfills essential technological functions. High sodium consumption has harmful health implications for cardiovascular and hypertension disorders. As a result, this study aimed to analyze the effect of KCl with Glycine and AMP on the physicochemical and sensory characteristics, purchase intent, and consumer perception of roasted chicken. NaCl/KCl replacement levels (0%, 25%, 50%, 75%, and 100%) were established, and consumer perception, liking, emotions, and purchase intent were evaluated. The different KCl levels, except for firmness, did not impact the physicochemical attributes. Even though higher replacement levels of KCl (75–100%) impacted chicken tenderness, it had no notable impact on panelists' liking scores and purchase intent. Health claims about the sodium content in roasted chicken have also been shown to significantly increase purchase intent, based on enhancing consumer's emotional responses. Regarding emotional responses, feelings of being unsafe and worried decreased their scores among the highest KCl replacement levels (75% and 100%). Positive emotional responses (feeling satisfied and pleased) were decisive consumer purchase intent predictors.

**Keywords:** roasted chicken; AMP; potassium chloride; glycine; sodium reduction

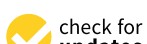



## 1. Introduction

The World Health Organization (WHO) indicates that high sodium intake has been associated with various diseases such as hypertension, cardiovascular disease, or stroke, so reducing sodium consumption can reduce blood pressure [1]. The latest data indicate that the world consumes much more sodium than necessary. In many cases, consumption far exceeds what is currently recommended by the WHO, which is 2 g of sodium (equivalent to 5 g of NaCl) per day. The Scientific Evaluation of Dietary Reference Intake (DRI) committee of the Washington Institute of Medicine/Food and Nutrition Commission reported that the adequate sodium intake is 1.5 g per day (3.8 g of salt per day), and the maximum tolerable level based on the prevention of blood increase is 2.3 g of sodium per

day. In turn, with increasing age, adequate intake decreases to 1.3 g per day of sodium for those over 50 and 1.2 g per day for those over 70 [2]. The current sodium intake worldwide exceeds the recommended intake in many countries, including the United States. The most significant contribution of sodium comes from processed foods. Excessive sodium consumption is associated with high blood pressure, which is a cardiovascular disease risk factor.

Most sodium comes from processed meat products, although there are differences between countries. The consumption of meat products is widespread, representing an essential part of the daily salt intake, which can reach 30%. In the USA, the daily contribution of salt from processed meat products is around 20% [3]. It must also be taken into account that not all meat products contain the same salt levels, and even within the same product, its salt content can vary considerably.

Reducing sodium salts poses an essential challenge for the meat industry, since sodium chloride (NaCl) fulfills essential technological functions, such as intervening in the extraction and solubilization of proteins [4]. Myofibrillar is necessary to improve properties such as water retention and texture [5]. NaCl also plays a fundamental role in developing the typical flavor of some meat products and ensuring the safety and extension of the useful life of most of them. Replacing sodium ions with potassium, calcium, and magnesium ions is feasible, even though imitating NaCl flavor is challenging [6]. Potassium chloride (KCl) has been used in roasted chicken [7]. The major flaw of adding KCl to chicken is that it causes a bitter aftertaste [8]. As a result, a suitable strategy to reduce sodium in chicken is to add bitter blockers such as AMP and Glycine to increase acceptability [9]. Statistical reports on the global poultry market indicate a growth of 4.1% between 2021 and 2025, with production reaching 100.9 million metric tons. Of this amount, production of nearly 13.4 million tons is expected, led by Brazil, the United States, and China [10].

Previous investigations have indicated that sodium reduction can affect food liking by altering sensory and physicochemical characteristics. They mainly involve sensory factors, since they contribute to the product's flavor characteristics [6,7]. However, more research is needed regarding the replacement of sodium in chicken. As a result, this study aimed to analyze the effect of KCl with Glycine and AMP on the physicochemical and sensory characteristics, purchase intent, and consumer perception of roasted chicken.

## 2. Material and Method

### 2.1. Ingredients and Roasted Chicken Preparation

The roasted chicken seasoning mix was made of granulated garlic (Mccormick, Baltimore, MD, USA) (38%), granulated onion (Mccormick, Baltimore, MD, USA) (32%), dried thyme (Mccormick, Baltimore, MD, USA) (6%), smoked paprika (Mccormick, Baltimore, MD, USA) (5%), mesquite powder (Mccormick, Baltimore, MD, USA) (9%), black pepper (Mccormick, Baltimore, MD, USA) (7%), and a salt-bitterness blockers mix (potassium chloride [Mccormick, Baltimore, MD, USA], NaCl [Norton, Chicago, IL, USA], glycine [USPFCC, J.T. Baker, Phillipsburg, NJ, USA], AMP [Adenosine-5'-monophosphate free acid, RPI, Mount Prospect, IL, USA]) (1.5%)), and ground rice hulls (Great American Spice Company, Rockford, MI, USA (1.5%) (Table 1). The addition of sodium chloride (NaCl), potassium chloride (KCl), Glycine, and 5' Adenosine Monophosphate (AMP) to the seasoning mix was optimized using a mixture design with space filling with constraints, consisting of concentrations, as illustrated in Table 1. A space-filling design applies design points throughout the design region and can adjust the linear constraints. The proportions of the salt mix were as follows: NaCl (0–100%), KCl (0–100%), Glycine (0–0.2%), and AMP (0–0.02%). The chicken thigh was provided from El Cortijo, Olancho, Honduras poultry farm. The roasted chicken was cooked by mixing two tablespoons of the seasoning mix per pound of chicken thigh. The product was heated in a MoTak convection oven (MoTak MCO-1 Single Full Size Convertible Gas Convection Oven, KaTom Drive Kodak, TN, USA) for 30 min at 200 °C [7].

**Table 1.** Salt and bitterness blocker mixture per formulation.

| Treatment * | NaCl% | KCl% | Glycine% | AMP% |
|---|---|---|---|---|
| **A** | 100 | 0 | 0 | 0 |
| **B** | 50 | 50 | 1 | 0.1 |
| **C** | 50 | 50 | 1 | 0.2 |
| **D** | 50 | 50 | 2 | 0.1 |
| **E** | 50 | 50 | 2 | 0.2 |
| **F** | 25 | 75 | 1 | 0.1 |
| **G** | 25 | 75 | 1 | 0.2 |
| **H** | 25 | 75 | 2 | 0.1 |
| **I** | 25 | 75 | 2 | 0.2 |
| **J** | 0 | 100 | 1 | 0.1 |
| **K** | 0 | 100 | 1 | 0.2 |
| **L** | 0 | 100 | 2 | 0.1 |
| **M** | 0 | 100 | 2 | 0.2 |

* Salt-bitterness blocker mix (KCl, NaCl, glycine, and AMP) is 1.5% *w/w* based on the weight of the seasoning mix.

## 2.2. Roasted Chicken's Physical/Technological Characteristics

Cooking loss was estimated by considering the weight of the chicken samples before and after the samples were prepared (N = 5) (N = number of tests performed per experiment). The cooking loss was estimated using Equation (1):

$$\text{Cooking loss} = (\text{Raw weight} - \text{cooked weight})/\text{Raw weight} \times 100 \qquad (1)$$

The chicken samples were examined with centrifugation (N = 10) for water-holding capacity (WHC), considering the weight before and after centrifugation. The WHC was estimated using Equation (2):

$$\text{WHC (\%)} = (\text{Wt2}/\text{Wt1}) \times 100 \qquad (2)$$

where Wt1 (before centrifugation) and Wt2 (after centrifugation) are the sample weights (g).

Color (L* (whiteness/darkness), a* (redness/greenness), and b* (yellowness/blueness)) were estimated with a colorimeter (Konica Minolta model CM-5, Japan) (N = 3). The firmness was analyzed (N = 10) using a texture analyzer (Stable Micro System Ltd., TA. X.T. plus, Surrey, England) with a shear probe (Warner-Bratzler knife) at a test speed of 200 mm/min [7]. Calibration was conducted with a height of 50 mm and a weight of 2 kg (N = 3).

## 2.3. Consumer Study

The sensory study was conducted in cubicles at UNAG (Catacamas, Olancho, Honduras). The study was conducted following the Declaration of Helsinki guidelines under the Food Research Ethics Committee supervised by the Honduran Association of Medicine and Nutrition (AS-ASHOMENU-0030-2022). Consumers were provided with a questionnaire (Compusense Inc., Guelph, ON, Canada). The panelists evaluated the product on the same day the samples were prepared. Consumers were provided with about half a pound of sample and a cup of water (Agua Azul, San Pedro, Honduras) with unsalted crackers (Nabisco, Northfield, IL, USA). Consumers (N = 325) were required to examine overall linking, flavor, aroma, color, tenderness, juiciness, bitterness, and saltiness on a 9-point hedonic scale. The panelists evaluated four samples. A Counter Balance Design was applied, resulting in 100 examinations per formulation. Demographic information was also gathered from the panelists. Purchase intent (P.I.) was considered by tasting on a "yes/no" scale. Emotion terms good, happy, interested, pleased, satisfied, unsafe, and worried (chosen based on the study made by Aleman et al. (2023) [7]) were evaluated on a 5-point scale. Consumers were provided with the subsequent health claims (H.C.): "KCl does not contain sodium, which is mainly associated with hypertension and heart diseases" (Table 2). The panelist evaluated O.L., emotion intensities, and P.I. again after receiving H.C. [11,12].

**Table 2.** Health claims for sodium content per formulation.

| F * | KCl% | Claim ** |
|---|---|---|
| **A** | **0%** **(Control)** | The chicken sample was made with identical sodium content to commercial roasted chicken. |
| **B–E** | **50%** | The chicken sample was prepared with 22–35% less sodium content than commercial roasted chicken. |
| **F–I** | **75%** | The chicken sample was prepared with 34–45% less sodium content than commercial roasted chicken. |
| **J–M** | **100%** | This product was prepared with 66–72% less sodium per serving than some commercial products. |

* Description refers to Table 1 for all formulations. ** Content claims were based on the FDA nutrient database.

### 2.4. Statistical Analysis

Statistical analysis was examined using SPSS 16 software (SPSS Inc., Chicago, IL, USA). One-way analysis of variance (ANOVA) and post hoc Tukey test ($\alpha = 0.5$) were applied to examine hedonic and emotional responses. Multivariate analysis of variance (MANOVA) and descriptive discriminant analysis (DDA) were applied to determine attributes accounting for overall product differences across samples regarding all sensory attributes at the same time. The two-related sample-dependent *t*-test was applied to study differences in O.L. and emotion terms before and after panelists acquired H.C. After H.C. was given, the McNemar test was applied to determine significant increases in P.I. Logistic regression analysis (LRA) was applied to estimate the impact of hedonic and emotional responses on P.I. likelihoods.

## 3. Results and Discussion

### 3.1. Physicochemical Results

The physicochemical properties (cook loss, firmness, WHC, and color) of the roasted samples are shown in Table 3. Including KCl, AMP, and glycine did not impact the chicken samples' cooking loss, WHC, or color (L*, a*, and b*). On the other hand, more tender chicken samples were observed in the treatments with 25–50% NaCl replacement levels, and the least tender chicken samples were observed in the treatments with 75–100% NaCl levels. Texture is one of the consumers' most essential sensory and quality characteristics [13]. Lee et al. [14] found that chicken samples with 75% to 100% KCl had increased hardness compared to samples with 0–50% KCl substitution. This tendency could be due to the reduced number sodium ions, which enhance protein solubility more than potassium ions. In the meat system, Na+ ions may also interact with positively charged myosin molecules, swelling the myofibrillar proteins in a more efficient way compared to K+ ions. NaCl is a water binder that enhances protein water-binding ability and boosts meat weight, making it softer [15].

**Table 3.** Physicochemical properties of roasted chicken.

| | **Means** | | | | | |
|---|---|---|---|---|---|---|
| **Trt *** | **L*** | **a*** | **b*** | **Firmness** | **Cook Loss (%)** | **Water Holding Capacity (%)** |
| **A** | 62.65 NS ** | 0.78 NS * | 25.54 NS * | 23.59 [b] | 33.04 NS * | 74.54 NS * |
| **B** | 62.94 | 0.72 | 24.65 | 21.02 [b] | 30.24 | 77.34 |
| **C** | 63.77 | 0.75 | 23.56 | 20.65 [b] | 32.82 | 75.76 |
| **D** | 62.45 | 0.70 | 25.69 | 21.78 [b] | 31.87 | 75.23 |
| **E** | 62.87 | 0.69 | 25.67 | 20.34 [b] | 30.93 | 77.34 |
| **F** | 62.89 | 0.59 | 24.67 | 27.19 [a] | 30.56 | 76.11 |
| **G** | 63.05 | 0.64 | 23.98 | 29.10 [a] | 31.76 | 75.09 |
| **H** | 62.35 | 0.69 | 24.96 | 30.21 [a] | 30.54 | 76.65 |
| **I** | 63.07 | 0.71 | 24.45 | 28.45 [a] | 32.45 | 77.05 |
| **J** | 63.06 | 0.77 | 25.67 | 29.34 [a] | 31.76 | 76.59 |

**Table 3.** *Cont.*

| | Means | | | | | |
|---|---|---|---|---|---|---|
| Trt * | L* | a* | b* | Firmness | Cook Loss (%) | Water Holding Capacity (%) |
| K | 62.79 | 0.65 | 25.60 | 30.56 [a] | 30.32 | 77.75 |
| L | 63.55 | 0.67 | 23.67 | 28.56 [a] | 30.48 | 76.32 |
| M | 63.16 | 0.76 | 24.20 | 31.56 [a] | 31.05 | 76.88 |
| Std. Error | 1.24 | 0.44 | 1.59 | 10.73 | 2.37 | 1.22 |

* Description refers to Table 1 for all formulations. ** Same letters are not significantly different ($p > 0.05$) within columns. NS: Not significant.

### 3.2. Consumer Study

The hedonic responses (overall liking, flavor, aroma, color, tenderness, juiciness, bitterness, and saltiness) are shown in Table 4. Typically, all sensory characteristics (hedonic responses) had scores greater than 4, with most reporting scores between 5 (neither liked nor disliked) and 6 (liked slightly).

**Table 4.** Liking scores for sensory properties of roasted chicken.

| Trt * | Color | Aroma | Flavor | Juiciness | Tenderness | Saltiness | Bitterness |
|---|---|---|---|---|---|---|---|
| A | 6.43 NS | 6.34 NS | 6.29 NS | 6.11 NS | 6.05 NS | 6.45 NS | 6.58 NS |
| B | 6.32 | 6.14 | 6.64 | 6.12 | 6.43 | 6.47 | 6.21 |
| C | 6.45 | 6.35 | 6.24 | 6.17 | 6.03 | 6.65 | 6.58 |
| D | 6.34 | 6.17 | 6.57 | 6.22 | 6.21 | 6.32 | 6.35 |
| E | 6.39 | 6.18 | 6.16 | 6.28 | 6.34 | 6.56 | 6.62 |
| F | 6.22 | 6.22 | 6.27 | 6.3 | 6.52 | 6.23 | 6.45 |
| G | 6.24 | 6.22 | 6.45 | 6.04 | 6.45 | 6.51 | 6.26 |
| H | 6.4 | 6.35 | 6.51 | 6.09 | 6.62 | 6.69 | 6.33 |
| I | 6.06 | 6.09 | 6.25 | 6.16 | 6.45 | 6.49 | 6.41 |
| J | 6.21 | 6.13 | 6.16 | 6.15 | 6.54 | 6.18 | 6.57 |
| K | 6.35 | 6.21 | 6.33 | 6.26 | 6.34 | 6.36 | 6.45 |
| L | 6.25 | 6.27 | 6.38 | 6.28 | 6.21 | 6.28 | 6.63 |
| M | 6.3 | 6.37 | 6.27 | 6.26 | 6.28 | 6.36 | 6.27 |
| Std. Error | 0.59 | 0.47 | 0.78 | 0.48 | 0.59 | 0.75 | 0.67 |

* Description refers to Table 1 for all formulations. NS: Not significant.

The treatments indicated no significant differences in overall liking, flavor, aroma, color, tenderness, juiciness, bitterness, and saltiness. Correspondingly, Lee et al. [13] did not find significant differences in sensory characteristics among treatments (samples with sodium-marinated levels within 25–100%). Nevertheless, other studies have shown that when replacing NaCl with potassium chloride, lower liking scores were observed when compared to NaCl samples in mayonnaise [16], roasted peanuts [17], frankfurter-type sausages [18], dipping sauces [19], and oil-in-water emulsions [20]. The greatest issue with substituting NaCl with other salts is that it can cause off flavors. In meat sausages, Bastianello et al. [21] reported lower scores for tenderness, aroma, bitterness, flavor, and saltiness in formulations with lower concentrations of NaCl. Selecting the proper sodium substitute is crucial to obtaining desired liking scores [22]. Lowering sodium in meat systems is challenging considering the difficulty to mimic sodium chloride's (NaCl) outstanding saltiness flavor [23].

### 3.3. Health Claim Impact on Overall Liking and Emotions

Differences in liking scores and emotional terms before and after health claims were provided to the panelists are shown in Table 5. All formulations were not significantly different ($p \geq 0.05$) in overall liking scores when comparing samples with 100% NaCl before or after the health claims were given to the panelists. The results of these observations

are related to the sensory liking scores of overall liking, flavor, aroma, color, tenderness, juiciness, bitterness, and saltiness, where none of the treatments were different from each other or from the control samples. Health claims may increase consumers' liking scores for chicken samples [7]. On the other hand, Verbeke [24] reported no significant differences ($p \geq 0.05$) in acceptance between samples with 100% NaCl and reduced sodium cheese. Low-salt products are not perceived as tasty as more salted alternatives in this group of consumers. Correspondingly, Verbeke [24] reported that informed consumers would lean more towards taste than health claim information in impacting liking scores.

**Table 5.** Liking and emotion scores before and after beneficial health claims.

| F * | Overall Liking | | Good | | Happy | | Pleased | | Satisfied | | Unsafe | | Worried | | Guilty | |
|---|---|---|---|---|---|---|---|---|---|---|---|---|---|---|---|---|
| **Time** | **Before** | **After** | **Before** | **After** | **Before** | **After** | **Before** | **After** | **Before** | **After** | **Before** | **After** | **Before** | **After** | **Before** | **After** |
| A | 5.56 ns | 5.54 NS | 2.23 ns | 2.47 NSϱ | 2.50 ns | 2.30 NS | 2.59 ns | 2.44 NS | 2.51 ns | 2.47 NS | 1.54 ns | 1.97 A | 1.47 ns | 1.98 A | 1.55 ns | 1.94 A |
| B | 5.34 | 5.55 | 2.65 | 2.76 | 2.67 | 2.44 | 2.49 | 2.65 | 2.67 | 2.63 | 1.67 | 1.92 AB | 1.55 | 1.67 B | 1.43 | 1.68 AB |
| C | 5.43 | 5.76 | 2.37 | 2.90 | 5.60 | 2.65 | 2.36 | 2.43 | 2.77 | 2.70 | 1.66 | 1.84 B | 1.60 | 1.55 B | 1.58 | 1.55 BC |
| D | 5.54 | 5.46 | 2.48 | 2.89 | 2.51 | 2.75 | 2.70 | 2.65 | 2.51 | 2.73 | 1.54 | 1.83 B | 1.65 | 1.50 B | 1.48 | 1.32 C |
| E | 5.41 | 5.39 | 2.66 | 2.65 | 2.51 | 2.68 | 2.46 | 2.40 | 2.65 | 2.68 | 1.50 | 1.80 B | 1.54 | 1.55 B | 1.60 | 1.32 BC |
| F | 5.47 | 5.41 | 2.47 | 2.77 | 2.58 | 2.60 | 2.44 | 2.64 | 2.58 | 2.64 | 1.36 | 1.77 B | 1.57 | 1.59 B | 1.47 | 1.37 BC |
| G | 5.55 | 5.36 | 2.57 | 2.69 | 2.57 | 2.59 | 2.48 | 2.58 | 2.59 | 2.70 | 1.58 | 1.75 B | 1.58 | 1.63 B | 1.53 | 1.39 BC |
| H | 5.35 | 5.47 | 2.48 | 2.65 | 2.50 | 2.48 | 2.37 | 2.59 | 2.60 | 2.59 | 1.63 | 1.80 B | 1.50 | 1.64 B | 1.58 | 1.33 BC |
| I | 5.48 | 5.55 | 2.59 | 2.80 | 2.47 | 2.60 | 2.63 | 2.72 | 2.71 | 2.66 | 1.44 | 1.82 B | 1.47 | 1.69 B | 1.55 | 1.16 C |
| J | 5.57 | 5.82 | 2.66 | 2.88 | 2.48 | 2.72 | 2.70 | 2.88 | 2.67 | 2.69 | 1.61 | 1.72 B | 1.50 | 1.45 B | 1.62 | 1.15 C |
| K | 5.44 | 5.54 | 2.58 | 2.70 | 2.43 | 2.57 | 2.54 | 2.80 | 2.54 | 2.58 | 1.60 | 1.76 B | 1.55 | 1.40 B | 1.44 | 1.15 C |
| L | 5.63 | 5.66 | 2.36 | 2.76 | 2.44 | 2.59 | 2.41 | 2.79 | 2.58 | 2.70 | 1.57 | 1.82 B | 1.59 | 1.48 B | 1.49 | 1.10 C |
| M | 5.58 | 5.60 | 2.60 | 2.83 | 2.38 | 2.69 | 2.46 | 2.71 | 2.59 | 2.77 | 1.39 | 1.80 B | 1.57 | 1.43 B | 1.46 | 1.06 C |
| **Std. Error** | 0.49 | 0.58 | 0.48 | 0.67 | 0.62 | 0.55 | 0.57 | 0.59 | 0.67 | 0.77 | 0.50 | 0.57 | 0.55 | 0.69 | 0.74 | 0.33 |

\* Description refers to Table 1 for all formulations. NS = not significant (After). ns = not significant (Before). ϱ means and letters are significantly different between before vs. after nutrient content claims.

Concerning positive emotions, a significant increase in the "satisfied" emotion was observed for a formulation containing 100% KCl, and decreasing scores for the "good" emotion were noted in control samples (100% NaCl). When examining negative emotions, the negative emotion "worried" had higher scores after the health claim for control samples, and a decrease in the "safe" emotion was noted in 75–100% NaCl samples. Furthermore, the "worried" emotion reported lower scores for treatments with 75 and 100% NaCl samples. Likewise, Aleman et al. [7] reported that negative emotions such as feeling "unsafe" decreased after the H.C. Emotional terms such as satisfaction, feeling unsafe, and worry are reported to be related to chicken [7,25,26]. Typically, liking scores of negative emotions decrease, and positive emotions increase after the H.Cs. [27].

*3.4. Purchase Intent of Chicken Samples*

The purchase intent of chicken samples, as affected by heath claims, is shown in Table 6. McNemar's test was used to examine the impact of health claims on purchase intent. Only samples with 75% and 100% sodium increased significantly in purchase intent. Similarly, Aleman et al. (2023) [7] reported that roasted chicken samples with 75% and 100% Na reduction significantly increased positive purchase intent. Furthermore, Bower et al. (2003) [28] evaluated that purchase intention of reduced-fat food products, and nutritional claims seemed to positively affect purchasing decisions. Carraro et al. (2012) [29] studied the acceptability of reduced sodium bologna sausages seasoned with spices and seasonings. Sausages that substituted 50% NaCl with KCl only, without incorporating herbs and spices, harmed purchase decisions. However, when spices and seasonings were incorporated into the products, buying intention improved. A suitable strategy to lower sodium in food products is to incorporate tastants like seasoning to enhance saltiness perception [30].

**Table 6.** Purchase intent of roasted chicken samples before and after sodium health claims.

| F * | mg Na/114 g Chicken Breast ^ | PIb (%) ** | PIa (%) | McNemar's Tests Statistic | Asymptotic PR > S | 95% Lower CL | 95% Upper CL |
|---|---|---|---|---|---|---|---|
| A | 135.189 | 45 | 42.003 | 0.9 | 0.288 | 0.666 | 0.891 |
| B | 104.949 | 37.503 | 42.003 | 2.7 | 0.13 | 0.711 | 0.9 |
| C | 104.949 | 49.5 | 52.497 | 0.9 | 0.288 | 0.666 | 0.891 |
| D | 104.949 | 52.497 | 54 | 0.126 | 0.63 | 0.531 | 0.837 |
| E | 104.949 | 36 | 38.997 | 0.603 | 0.369 | 0.576 | 0.855 |
| F | 89.829 | 46.503 | 46.503 | 0 | 0.9 | 0.585 | 0.855 |
| G | 89.829 | 46.503 | 47.997 | 0.297 | 0.504 | 0.711 | 0.9 |
| H | 89.829 | 40.5 | 45 | 2.7 | 0.07 *** | 0.711 | 0.9 |
| I | 89.829 | 40.5 | 46.503 | 3.6 | 0.08 *** | 0.666 | 0.891 |
| J | 45.9 | 45 | 52.497 | 3.213 | 0.19 | 0.549 | 0.837 |
| K | 45.9 | 42.003 | 43.497 | 0.9 | 0.288 | 0.81 | 0.9 |
| L | 45.9 | 51.003 | 55.503 | 1.161 | 0.234 | 0.531 | 0.828 |
| M | 45.9 | 42.003 | 45 | 1.8 | 0.144 | 0.756 | 0.9 |

* Description refers to Table 1 for all formulations. ** PIb: purchase intent before; PIa: purchase intent after. *** Not statistically significant at $\alpha = 0.05$ but potentially significant in cases of increasing significance levels to $\alpha = 0.1$. ^ By calculation.

### 3.5. Predicting Purchase Intent Using Logistic Regression Analysis (LRA)

The LRA was conducted to forecast the formulations' likelihood of influencing purchase intent based on sensory characteristics, demographic information, and emotional terms (Table 7). The results showed that the sensory characteristics of saltiness and flavor were meaningful predictors of buying decisions even before delivering the health claim. Likewise, other examinations have demonstrated that H.C. positively impacted P.I. in chicken and bologna sausages [31] and roasted chicken [7]. However, buyers may consume more based on pricing and product safety over health claims when considering purchase preferences for poultry products [32]. On the other hand, Kim et al. [33] considered panelists' understanding of sodium labeling and found that sodium was perceived as unrelated to concerns about heart illness or bone health compared to kidney illness in consumers' perception. In other words, prior understanding of the likely threats of high sodium consumption did not affect the purchase decision of non-salt customers. This study reveals interesting findings, since the purchase intent analysis based on logistic regression analysis suggests that sodium content claims rely more on perspectives regarding health claims than on demographic information.

**Table 7.** Odds ratio for predicting purchase intent before and after sodium nutrient content claims by logistic regression procedure.

| | Before * | | After | |
|---|---|---|---|---|
| Variables | Pr > ChiSq ** | Odds Ratio | Pr > ChiSq | Odds Ratio |
| Gender | 0.858325 | 0.9766 | 0.21 | 1.2654 |
| Lower sodium | 0.23275 | 1.36 | 0.45 | 0.75 |
| Salt user | **0.03154** | 0.48 | **0.01** | 0.34 |
| Color | 0.50125 | 1.01 | - | - |
| Aroma | 0.33706 | 0.86 | - | - |
| Flavor | 0.46509 | 1.33 | - | - |
| Tenderness | 0.12426 | 0.81 | - | - |
| Juiciness | 0.59145 | 0.99 | - | - |
| Saltiness | 0.07935 | 1.18 | - | - |
| Bitterness | 0.41705 | 1.02 | - | - |
| Overall liking | **<0.0001** | 2.37 | **<0.0001** | 3.99 |
| Good | 0.85825 | 0.92 | 0.23 | 0.72 |
| Happy | 0.23735 | 1.20 | 0.09 | 1.48 |
| Pleased | 0.77938 | 0.90 | 0.91 | 0.96 |

**Table 7.** *Cont.*

| Variables | Before * | | After | |
| | Pr > ChiSq ** | Odds Ratio | Pr > ChiSq | Odds Ratio |
|---|---|---|---|---|
| **Satisfied** | **<0.0001** | 2.15 | **<0.0001** | 2.23 |
| **Unsafe** | 0.74176 | 0.88 | 0.63 | 1.12 |
| **Worried** | 0.15390 | 0.60 | 0.51 | 0.74 |
| **Guilty** | 0.37772 | 1.19 | 0.50 | 0.81 |

* Purchase intent before and after content claims were presented to consumers. ** Statistically significant *p*-values in bold print ($p < 0.05$).

### 3.6. Liking Using MANOVA

Liking, as affected by sensory characteristics, demographic information, and emotional terms, was examined using MANOVA and DDA (Table 8). According to the Pillai's Trace and Wilk's Lambda, it was lower than the significance value ($p < 0.05$). When conceding health claims, emotional terms like "unsafe" and "worried" were the main factors that impacted the overall liking of roasted chicken. Health claims may impact customers' emotions. Aleman et al. [7] concluded that the negative emotional terms of "worried" and "unsafe" declined after H.C., since panelists could be worried about sodium in chicken, especially non-salt customers. Incorporating bitterness blockers AMP and glycine into 100% sodium-substituted roasted chicken with KCl yields similar sensory perceptions to controlling metallic and bitter flavors. However, it is not associated with impacting the product liking scores. Sodium claims in reaction to sodium replacement did not improve consumer attention when considering overall liking regarding sodium's health issues. Salt substitutes (SSBS) are products with less sodium than common salt. The amounts of sodium in SSBS are reduced by replacing some of the sodium with potassium or other minerals. SSBS could reduce the risks of using table salt since consuming too much sodium and too little potassium contributes to high blood pressure. Worldwide, high blood pressure is the largest cause of preventable deaths, especially because it causes stroke, acute coronary syndrome (ACS, where less blood reaches the heart), and kidney problems. Significant action was taken to reduce sodium intake; nevertheless, reformulation alone will not suffice to replace sodium with salt substitutes. For future research, studying the consumer perception of the health benefits of reducing sodium to a broader demographic is encouraging. Educating the consumer should be a priority in increasing the acceptance of low or free-sodium products.

**Table 8.** Canonical structure describing group differences among low-sodium roasted chicken.

| | Canonical Structure | | |
| Variable | Can 1 | Can 2 | Can 3 |
|---|---|---|---|
| **Color** | 0.326 | 0.112 | 0.270 |
| **Aroma** | 0.162 | 0.227 | 0.113 |
| **Flavor** | 0.050 | −0.053 | 0.125 |
| **Tenderness** | 0.078 | 0.156 | 0.024 |
| **Juiciness** | −0.013 | 0.065 | −0.130 |
| **Saltinnes** | 0.174 | −0.011 | 0.182 |
| **Bitterness** | 0.074 | −0.007 | 0.179 |
| **Good [B]** | −0.007 | 0.049 | −0.001 |
| **Happy** | 0.036 | 0.030 | −0.102 |
| **Pleased** | 0.010 | 0.082 | −0.142 |
| **Satisfied** | −0.001 | −0.035 | 0.041 |
| **Unsafe** | 0.042 | −0.006 | 0.037 |
| **Worried** | 0.046 | 0.106 | 0.026 |
| **Guilty** | 0.156 | 0.196 | −0.041 |
| **Overall liking** | 0.085 | 0.038 | 0.106 |
| **Good [A]** | −0.234 | 0.214 | 0.073 |
| **Happy** | −0.126 | 0.053 | 0.076 |

**Table 8.** *Cont.*

| | Canonical Structure | | |
|---|---|---|---|
| Variable | Can 1 | Can 2 | Can 3 |
| Pleased | −0.129 | 0.191 | 0.158 |
| Satisfied | −0.068 | 0.070 | 0.174 |
| Guilty | 0.390 | 0.026 | −0.242 |
| Unsafe | 0.315 | −0.002 | −0.210 |
| Worried | 0.330 | 0.115 | −0.201 |
| Overall liking | 0.001 | 0.144 | 0.186 |
| Cumulative variance | 0.29 | 0.44 | 0.59 |
| Wilk's Lambda *p* value | | 0.0111 | |

Based on pooled within-group variances. [B] Before nutrient content claims were presented. [A] After nutrient content claims were presented.

## 4. Conclusions

This study aimed to analyze the effect of KCl with Glycine and AMP on the physicochemical and sensory characteristics, purchase intent, and consumer perception of roasted chicken. The physicochemical characteristics were generally not affected by different KCl replacements, except for firmness. Even though the higher replacement of KCl levels (75–100%) impacted the chicken's tenderness, it had no notable impact on panelist liking scores and purchase intent. Health claims about the sodium content in roasted chicken have also been shown to significantly increase purchase intent by enhancing consumer emotional response. For future studies, exploring consumer behavior regarding the health benefits of reducing sodium across a broader demographic is recommended. Also, examining acceptance when educating the consumer should be an emphasis to improve purchase intent and liking of low- or free-sodium products.

**Author Contributions:** Conceptualization, R.S.A.; methodology, R.S.A.; software, R.S.A.; formal analysis, R.S.A. (most of the research), J.A.M.F., M.d.J.Á.G. and A.Y.; resources, R.S.A., H.Z.F., I.M.-F. and J.A.M.F.; data curation, R.S.A.; writing—original draft preparation, R.S.A.; writing—review and editing, I.M.-F., R.S.A. and D.M.-V.; project administration, R.S.A. and J.A.M.F.; funding acquisition, R.S.A. and J.A.M.F. All authors have read and agreed to the published version of the manuscript.

**Funding:** This research was funded by the University National of Agriculture (Honduras) (Ref. HnSubvención 003-2023) with the International Development Research Center of Canada (IDRC) and the General Secretariat of the Council Central American University Superior (CSUCA) (Ref. C-DSIP-008-2023-UNAG).

**Data Availability Statement:** Data are contained within the article.

**Conflicts of Interest:** The authors declare no conflicts of interest.

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
