# Peer review of "Impact of 5’ Adenosine Monophosphate, Potassium Chloride, and Glycine on the Physicochemical and Sensory Characteristics of Sodium-Reduced Chicken"

_2674-0311, doi:10.3390/dietetics3020008_

Round 1
Reviewer 1 Report
Comments and Suggestions for Authors
The manuscript submitted to your journal falls within the aims and scope and has driven an important hypothesis, that is the substitution of sodium chloride by potassium chloride, AMP, and glycine. The manuscript is in within the field of nutrition and is important for the consumers to get more knowledge about the impact of sodium chloride in cooked meat (chicken) concerning their health balance.
The manuscript is in general well prepared and statistical analysis has been implemented to highlight better the findings of the study. I have included in the attacghed pdf the corrections and clarifications the authors should provide in order to improve further the quality of their study.
Based on my overall omments, I suggest a minor revision prior to further consideration for publication.

The English language is in good level, but it requires some revision. See attached file.
Author Response
The manuscript submitted to your journal falls within the aims and scope and has driven an important hypothesis, that is the substitution of sodium chloride by potassium chloride, AMP, and glycine. The manuscript is in within the field of nutrition and is important for the consumers to get more knowledge about the impact of sodium chloride in cooked meat (chicken) concerning their health balance.
The manuscript is in general well prepared and statistical analysis has been implemented to highlight better the findings of the study. I have included in the attached pdf the corrections and clarifications the authors should provide in order to improve further the quality of their study.
Based on my overall comments, I suggest a minor revision prior to further consideration for publication.
Comments on the Quality of English Language
The English language is in good level, but it requires some revision. See attached file.
Answer: Recommendation were taken into consideration.
Reviewer 2 Report
Comments and Suggestions for Authors
After reviewing the manuscript titled "Impact of 5' Adenosine Monophosphate Potassium Chloride Glycine on the Physicochemical and Sensory Characteristics of Sodium Reduced Chicken," here are some specific comments:
- The study presents a significant investigation into sodium reduction in chicken, addressing a crucial public health concern. The utilization of 5' AMP and glycine as bitterness blockers is an innovative approach. However, the manuscript could further discuss the novelty of this approach compared to existing literature.
- The abstract needs to be improved.
- The objective needs to rewrite and elaborate.
- The methodology section is detailed, providing clear insight into the experimental design, ingredients used, and statistical analysis. To enhance clarity and replicability, consider specifying any assumptions made during the analysis and providing more detail on the mixture design optimization.
- The results are comprehensively presented. However, the discussion on the implications of these results on food science and public health could be expanded. Specifically, exploring the sensory acceptance of reduced sodium products and its potential impact on consumer behaviour would be valuable.
- The conclusion needs to be improved with some recommendation for further study.
- Whereas possible, use error bars in the figures and standard deviation in the results (numbers).
Comments on the Quality of English Language- Revision is necessary.
Author Response
After reviewing the manuscript titled "Impact of 5' Adenosine Monophosphate Potassium Chloride Glycine on the Physicochemical and Sensory Characteristics of Sodium Reduced Chicken," here are some specific comments:
- The study presents a significant investigation into sodium reduction in chicken, addressing a crucial public health concern. The utilization of 5' AMP and glycine as bitterness blockers is an innovative approach. However, the manuscript could further discuss the novelty of this approach compared to existing literature.
Answer: Introduction was improved.
- The abstract needs to be improved.
Answer: Abstract was improved.
- The objective needs to rewrite and elaborate.
Answer: Objective was rewritten.
- The methodology section is detailed, providing clear insight into the experimental design, ingredients used, and statistical analysis. To enhance clarity and replicability, consider specifying any assumptions made during the analysis and providing more detail on the mixture design optimization.
Answer: mixture design optimization was detailed.
- The results are comprehensively presented. However, the discussion on the implications of these results on food science and public health could be expanded. Specifically, exploring the sensory acceptance of reduced sodium products and its potential impact on consumer behavior would be valuable.
Answer: Discussion was added on exploring the sensory acceptance of reduced sodium products and its potential impact on consumer behavior would be valuable.
- The conclusion needs to be improved with some recommendation for further study.
Conclusions
Answer: conclusion was improved.
Round 2
Reviewer 2 Report
Comments and Suggestions for Authors
The revised manuscript is much improved however, we still need to revise the abstract, objective of the study, and conclusion. After minor revision, the manuscript can be accepted.
Comments on the Quality of English LanguageNeet to be improved.
Author Response
Abstract and conclusion was improved.
